# Disseminated Fungal Infection and Fungemia Caused by *Trichosporon asahii* in a Captive Plumed Basilisk (*Basiliscus plumifrons*)

**DOI:** 10.3390/jof7121003

**Published:** 2021-11-24

**Authors:** Chieh Lo, Chu-Lin Kang, Pei-Lun Sun, Pin-Huan Yu, Wen-Ta Li

**Affiliations:** 1Xpark (Taiwan Yokohama Hakkeijima Inc.), Taoyuan 32056, Taiwan; chieh.lo@xpark.com.tw (C.L.); k5987723@gmail.com (C.-L.K.); 2Research Laboratory of Medical Mycology, Department of Dermatology, Chang Gung Memorial Hospital, Linkou Branch, Taoyuan 33305, Taiwan; sunfungus@gmail.com; 3College of Medicine, Chang Gung University, Taoyuan 33302, Taiwan; 4Institute of Veterinary Clinical Sciences, National Taiwan University, Taipei 10617, Taiwan; pinhuan@ntu.edu.tw; 5National Taiwan University Veterinary Hospital, Taipei 10672, Taiwan; 6Pangolin International Biomedical Consultant Ltd., Keelung 20145, Taiwan

**Keywords:** formalin-fixed paraffin-embedded (FFPE), fungemia, histopathology, plumed basilisk (*Basiliscus plumifrons*), polymerase chain reaction (PCR), *Trichosporon asahii*

## Abstract

*Trichosporon* spp. are heavily arthroconidiating fungi and widely distributed in nature. Due to the similar fungal morphology, confusion among *Trichosporon* spp., *Geotrichum* spp., and *Nannizziopsis* spp. in reptiles is apparent and cannot be overlooked. Although few reptile *Trichosporon* isolates have been examined using the newer speciation criteria, the information on *Trichosporon asahii* in reptiles is still scarce. In the present study, we report the case of disseminated fungal infection and fungemia caused by *T. asahii* in a captive plumed basilisk (*Basiliscus plumifrons*). Multiple 0.2–0.5 cm, irregularly shaped, ulcerative nodules on the left hind foot were observed. The animal died due to the non-responsiveness to treatment. A microscopic evaluation revealed the fungal infection that primarily affected the left hind foot and right lung lobe with fungal embolisms in the lung and liver. The molecular identification of the fungal species by the DNA sequences of the ITS regions and D1/D2 gene from the fungal culture and ITS regions, from formalin-fixed paraffin-embedded (FFPE) lung tissues, were completely matched to those of *T. asahii*. The current report describes the first confirmed case of disseminated fungal infection and fungemia caused by *T. asahii* in a captive plumed basilisk.

## 1. Introduction

*Trichosporon* spp. are basidiomycetous, yeast-like, and heavily arthroconidiating fungi that are widely distributed in nature and found predominantly in tropical and temperate areas [1,2]. They are normal microbiota present in soil and can colonize the alimentary tract, respiratory tract, and skin of humans [2,3]. *Trichosporon* spp. have been identified as opportunistic pathogens causing systemic and life-threatening infections, especially in immunosuppressed patients [1,2]. In reptiles, *Trichosporon* spp. have been cultured from the intestine and feces of healthy and diseased reptiles [3,4,5] and reported to cause dermatitis in a North African spiny-tailed lizard (*Uromastyx acanthinura*) and a Hermann’s tortoise (*Testudo hermanni*); pulmonary nodules and intestinal ulcers an aquatic South American snake; hepatic and renal nodules in three captive banded rock rattlesnakes (*Crotalus lepidus klauberi*); keratoconjunctivitis in a leopard gecko (*Eublepharis macularius*); and oral lesions in an Aldabra giant tortoise (*Aldabrachelys gigantea*), a Nile crocodile (*Crocodylus niloticus*), and a spectacled caiman (*Caiman crocodilus*) [3,6,7,8,9]. 

The *Trichosporon* genus has undergone extensive revision and the species *T. beigelii* has been replaced by several species [1,2,10]. However, only few reptile *Trichosporon* isolates have been identified using the newer speciation criteria [3,9]. In a previous survey of 127 reptiles, 4 cutaneous *Trichosporon* isolates from 2 green iguanas (*Iguana iguana*), a blood python (*Python curtus*), and a ball python (*Python regius*) were identified as *T. asahii*, but no detailed information on the clinical presentation and pathological findings were provided [4]. In addition, due to the similar fungal morphology, confusion among *Trichosporon* spp., *Geotrichum* spp., and *Nannizziopsis* spp. is possible. In the present study, we reported a case of disseminated fungal infection and fungemia caused by *T. asahii* in a captive plumed basilisk (*Basiliscus plumifrons*), which can provide information on a more accurate diagnosis of *T. asahii* infection.

## 2. Materials and Methods

### 2.1. Case

A captive, adult, male, plumed basilisk was housed in Xpark (Taiwan Yokohama Hakkeijima Inc.). The lizard was kept in a 145 × 100 × 120 cm glass tank with soil and running water for 5 months. The lizard showed anorexia and multiple 0.2–0.5 cm, irregularly shaped, ulcerative nodules on the left hind foot were observed (Figure 1). Dorsoventral (DV) vertical radiographic views of the left hind foot demonstrated that nodules on the second to third phalanges of the second digit and second phalanx of the third digit had an increased opacity with bone lysis (Figure 2). The radiographic appearance of the right lung lobe revealed mild, irregularly shaped foci of increased opacity, which were interpreted as pulmonary nodules or an overlapping of the heart. No significant findings were detected in the other organ systems. Lateral horizontal radiographic views, which were usually recommended to evaluate the lung in reptiles, were not performed due to the animal’s condition. Antibiotic treatment (ceftazidime: 5 mg/kg SC q48h for 7 days) was given, but the lizard died due to the non-responsiveness to the treatment. 

A complete necropsy with standardized procedures was performed. Gross examination revealed a diffuse consolidated right lung lobe with multiple variably sized grey-to-tan nodules. No significant findings were detected in the left lung lobe and other internal organs. Representative tissue samples, including the brain, foot, and plug (heart; gastrointestinal tract; liver; pancreas; spleen; kidneys; and gonads), were collected, fixed in 10% neutral buffered formalin, and submitted to the Pangolin International Biomedical Consultant Ltd. for pathological examination. Fresh tissue samples from the digital nodules were submitted for fungal culture with a subsequent mycological analysis.

### 2.2. Histological Examination

The formalin-fixed tissue samples were trimmed, dehydrated through a series of graded ethanol, and then infiltrated with paraffin. The formalin-fixed paraffin-embedded tissues were s-sectioned at 4 µm and stained with hematoxylin and eosin (H&E), periodic acid-Schiff (PAS), Jones’ methenamine silver (JMS), Brown and Brenn (B&B) Gram, and Ziehl–Neelsen (ZN) stains. All histological slides were scanned using a PANNORAMIC DESK II DW Slide Scanner (Budapest, Hungary). A microscopic examination was performed using certified 3DHistech CaseViewer software 2.3.0.99276 (Budapest, Hungary).

### 2.3. Fungal Species Identification

Fresh samples from the digital nodule were obtained and inoculated on inhibitory mold agar with chloramphenicol and gentamicin (ICG™) (Creative CMP^®^, Taiwan) and Mycosel™ agar (BD Difco™, Detroit, MI, USA), and then incubated at 25 ℃. The fungi grown from the primary culture were used for subsequent molecular identification. 

Polymerase chain reactions (PCRs) using primer sets, targeting internal transcribed spacer (ITS) regions and the partial 28S ribosomal DNA (D1/D2) gene, were performed with subsequent DNA sequencing [11,12]. In order to identify the fungal species affecting the lung, a PCR with subsequent DNA sequencing using formalin-fixed paraffin-embedded (FFPE) lung tissues was per-formed. DNA was extracted from 80 µm sections of tissue using the QIAamp DNA FFPE Tissue Kit (Qiagen, Valencia, CA, USA) and used as PCR templates [13]. Primer sets targeting the ITS regions of *T. asahii* (TAF: 5′-CGC ATC GAT GAA GAA CGC AG-3′ and TAR: 5′-GCG GGT AGT CCT ACC TGA TT-3′) designed by using Primer3 (http://www.ncbi.nlm.nih.gov/tools/primer-blast/, 1 September 2021) were used. The obtained DNA sequences were compared with DNA sequences available in GenBank using the Basic Local Alignment Search Tool (BLAST) server from the National Center for Biotechnology Information.

## 3. Results

### 3.1. Histopathology

In the left hind foot, the dermis and hypodermis were extensively expanded and infiltrated to the underlying phalangeal bones that were variably sized, discrete to coalescing granulomas, composed of a central area of eosinophilic cellular debris admixed with mixed inflammatory cells surrounded by epithelioid macrophage and multinucleated giant cells rimmed by concentric layers of fibrous connective tissue (Figure 3A). Large amounts of erythrocytes, necrotic cell debris, and mixed inflammatory cells with varying numbers of fungal elements were multifocally filling the central lumen and faveolar spaces of the right lung lobe (Figure 3B). Small-to-large sized blood vessels from the lung and liver were occluded by thromboembolism composed of fibrin, necrotic cell debris, degenerate mixed inflammatory cells, and variable numbers of fungal hyphae and arthroconidia (Figure 3C–D). Under PAS and JMS staining, it was observed that scattered throughout the granulomas and thromboembolism were variable numbers of fungal hyphae that were PAS- and JMS-positive, branched, septate, and approximately 2 to 4.5 μm in diameter with rare arthroconidia (Figure 3E–F). No evidence of infectious microorganism was detected under the B&B Gram and ZN stains.

### 3.2. Fungal Identification

The DNA sequences of the ITS regions (accession numbers: OL468573) and D1/D2 gene (accession numbers: OL468574), from the fungal culture, and ITS regions (accession number: OL441035), from FFPE lung tissues, were completely matched to those of *T. asahii*.

## 4. Discussion

The microscopic evaluation of the submitted tissues revealed a disseminated fungal infection that primarily affected the left hind foot and right lung lobe with fungal embolisms in the lung and liver. These findings indicate that the current fungi are angioinvasive and cause disseminated lesions in multiple organs. Furthermore, the identification of fungal species by molecular analysis through PCR and DNA sequencing is performed by the DNA extracted from not only the isolated fungal colony of foot lesions, but also the FFPE lung tissue. The results of DNA sequencing demonstrated that the fungi observed in the foot lesions and lung were *T. asahii*. The results of histomorphology, PCR, and DNA sequencing confirmed the final diagnosis of disseminated fungal infection and fungemia caused by *T. asahii*. 

Dermatomycosis is a term referring to the fungal infection of the skin and cutaneous adnexa, and the stages of dermatomycosis can be classified as superficial, deep, and disseminated, based on the invasiveness of the fungi [11,12]. Systemic mycosis can occur with or without concurrent dermatomycosis, and severe dermatomycosis can progress to systemic mycosis [3,14]. It is a common that systemic mycosis initially involves the lungs, and it is either confined to the lungs or disseminated to other organ systems [3,14]. The spreading of fungal elements in the visceral organs can occur hematogenously, or by a direct infiltration through the serosa and coelomic membranes to the adjacent organs [3,14]. Pneumonia, caused by fungal infection, is relatively common in tortoises and crocodilians, but it has been reported in a variety of reptile species [3]. Although the contributing factors, including the thermotolerance of the fungi, host immune status, and host species, for establishing a fungal infection have been investigated by studies conducted in humans and plants, such factors are largely unexplored in reptiles [9,15]. Nevertheless, deep dermatomycosis and disseminated fungal infections have been reported in stressed and immunocompromised reptiles due to long distance transportation and suboptimal environmental conditions [9,15].

In this case, although fungemia and systemic spreading have been observed by the histopathology, the primary location and mechanism for *T. asahii* to establish the initial infection with subsequent systemic spreading is still undetermined. The possible mechanisms include, but are not limited to: (1) aspiration pneumonia in the right lung lobe with subsequent angioinvasion and spreading to the foot; (2) a fungal infection on the skin on the left hind foot with subsequent bone invasion, angioinvasion, and spreading to the lung; and (3) a fungal infection, simultaneously, in the foot and lung with subsequent angioinvasion. Since only the right lung lobe was affected by *T. asahii*, and the lesion was more consistent with bronchopneumonia rather than interstitial pneumonia, a diagnosis of aspiration pneumonia rather than a disseminated fungal infection affecting the lung is more likely. Aspiration pneumonia refers to pneumonia caused by the aspiration of foreign materials into the lungs through the airways [16]. The mechanism of fungal spreading, from the lung to the foot in this case, may be similar to that of feline lung–digit syndrome (FLDS), which is a syndrome used to describe the clinical progression of cutaneous metastases on digits metastasized from a primary pulmonary carcinoma [17]. In such a condition, multiple digits are commonly affected, which are consistent with our case. Therefore, although other possibilities cannot be completely excluded, aspiration pneumonia with a subsequent systemic spreading is the most likely underlying mechanism for this case.

In addition to the gross and histopathological findings, radiographic examination can be worthwhile to identify disseminated fungal infection in reptiles. In general, lizard lungs are more easily evaluated by the lateral horizontal radiographic views [18]. The utility of the DV vertical view is usually not satisfied due to the overlapping of the lung with the other coelomic organs (such as the heart and liver), but it is the only method that can be employed to evaluate bilateral lung lobes individually [18]. Based on the experience from this case, DV vertical and lateral horizontal radiographic views of the lung and other internal organs may be considered if there is a suspicious fungal infection on the feet and digits. 

In the present study, the current captive environment was recently established (i.e., a new facility), and the lizard was introduced from another facility. There are several possible contributing factors that can compromise the immune function and lead to a disseminated fungal infection, including transportation, environmental change, and different housemates. For the source of the current *T. asahii*, since *T. asahii* is widely distributed in the environment, the lizard in this study can be infected in the new facility. However, it is also possible that the lizard in the present study was infected by *T. asahii* in the previous facility (an old facility) as a subclinical/latent infection with the subsequent development of the disseminated infection in the current facility.

In conclusion, the current report describes the first confirmed case of disseminated fungal infection and fungemia caused by *T. asahii* in a captive plumed basilisk and discusses the underlying mechanism of fungal spreading. This case report can provide information for the accurate diagnosis of a disseminated fungal infection, which is important to gain a better understanding of the pathogenesis of fungal infections and to improve the animal welfare of reptiles in captivity.

## Figures and Tables

**Figure 1 jof-07-01003-f001:**
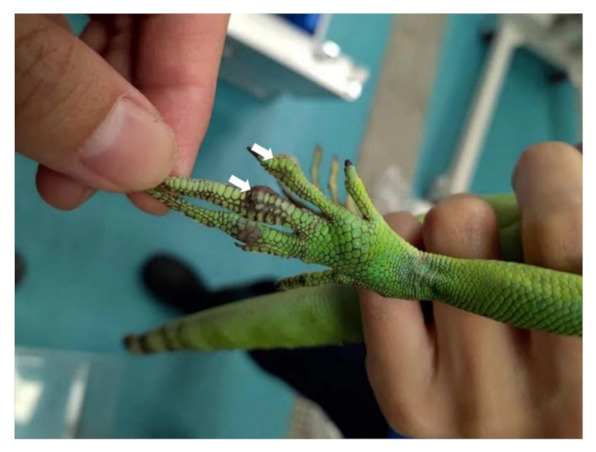
Two 0.2–0.5 cm ulcerative skin nodules on the left hind foot (arrows).

**Figure 2 jof-07-01003-f002:**
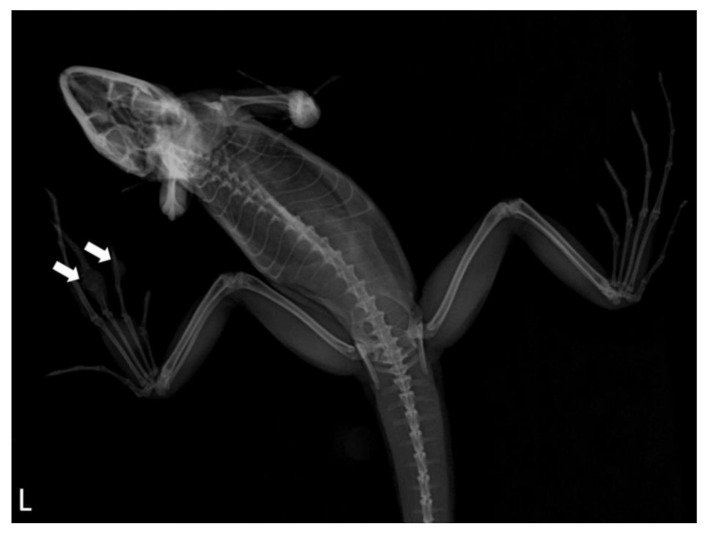
Dorsoventral (DV) vertical radiographic views of the left hind foot. Note the soft tissue mass on the digit with bone lysis (arrows).

**Figure 3 jof-07-01003-f003:**
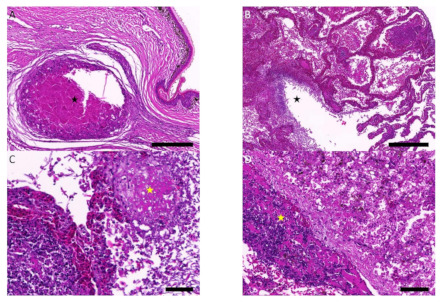
Histopathology. (**A**) Granuloma (☆) composed of a central area of eosinophilic cellular debris admixed with mixed inflammatory cells in the dermis; hematoxylin and eosin (HE) stain; and scale bar = 500 μm. (**B**) Central lumen and faveolar spaces (☆) of the right lung lobe filled with large amounts of erythrocytes, necrotic cell debris, and mixed inflammatory cells; HE stain; and scale bar = 500 μm. (**C**) Thromboembolism with fungal elements (☆) in the lung; HE stain; And scale bar = 50 μm. (**D**) Thromboembolism with fungal elements (☆) in the liver; HE stain; and scale bar = 100 μm. (**E**) Branched, septate fungal hyphae with rare arthroconidia; periodic acid-Schiff (PAS) stain; and scale bar = 50 μm. (**F**) Branched, septate fungal hyphae with rare arthroconidia; Jones’ methenamine silver (JMS) stain; and scale bar = 50 μm.

## Data Availability

The data presented in this study are available on request from the corresponding author. The data are not publicly available due to large file size without sustainable storage space.

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
