# Peer review of "Disseminated Fungal Infection and Fungemia Caused by Trichosporon asahii in a Captive Plumed Basilisk (Basiliscus plumifrons)"

_jof, 2021, doi:10.3390/jof7121003_

Round 1
Reviewer 1 Report
In results add micro and macroscopic information on the growth of the fungus. There is no information on this in the results and the DNA sequences really should work as confirmatory evidence (add photos).
Authors must deposit the sequence in the Gene bank , have their access number of the strain, and the percentage of homology.
Author Response
Dear reviewers:
The attached file is revised manuscript with tracked changes (jof-1475686) and without tracked changes (jof-1475686-clean). In the revised manuscript, we have made the following changes according to the reviewers’ comments.
Reviewer-1’s comments
- In results add micro and macroscopic information on the growth of the fungus. There is no information on this in the results and the DNA sequences really should work as confirmatory evidence (add photos). Authors must deposit the sequence in the Gene bank, have their access number of the strain, and the percentage of homology.
Response: Thanks for the comment. We have deposited the DNA sequencings in the Gene bank with accessory no. OL441035, OL468573, and OL468574. All the information has been updated in the manuscript (lines 136-138). For the macroscopic and microscopic features of the fungal culture: We sent the tissues to a diagnostic lab of a hospital for fungal culture, and they did not do the morphological studies of the fungal culture due to the manpower shortage in covid-19 pandemic. We feel very sorry that we don’t have the morphological information on the fungal culture.
We thank you and the reviewers very much for providing us so many valuable suggestions. We hope the revise that we have made in the revision meets the reviewer’s and your requests. We are looking forward to having our revised manuscript published in your distinguished journal ‘‘Journal of Fungi”
Sincerely yours,
Wen-Ta Li, DVM, PhD
Pangolin International Biomedical Consultant Ltd.
Email: heerolee1104@gmail.com
Reviewer 2 Report
This is a nice paper where the authors described the first confirmed case of disseminated fungal infection and fungemia caused by T. asahii in a captive plumed basilisk. Paper is well written and the invasive infection by T asahii was accurately documented.
Please, find bellow my main suggestions to the authors:
- Taxonomy of Trichosporon spp was updated and the authors should add an appropriate reference reviewing this aspect in their introduction. Most clinically important Trichosporon spp. such as T. asteroides, T. coremiiforme, T. dohaense, T. faecale, T. japonicum, T. inkin, T. ovoides, and especially T. asahii are still classified as Trichosporon. However, other relevant species able to cause infection in humans, including T. cutaneum, T. jirovecii, T. dermatis, T. mucoides and T. debeurmannianum now belong to Cutaneotrichosporon, whereas T. domesticum, T. loubieri, T. montevideense and T. mycotoxinivorans are classified in Apiotrichum (Liu et al. 2015a; Takashima et al. 2018; 2019; Aliyu et al. 2020; Li et al. 2020). In addition, species formerly known as T. pullulans, which belongs to Cystofilobasidiales, and is phylogenetically distinct from Trichosporonales now has been reclassified as Guehomyces pullulans (Fell and Scorzetti 2004) and, more recently, as Tausonia pullulans (Liu et al. 2015a). Of note, all those modifications were not mentioned in any of the references listed by the authors in the present paper. This topic was extensively reviewed by Amir Arastehfar et al, in their publication at Crit Rev Microbiol 2021 47(6):679-698. I suggest to incorporate this reference right after the following sentence of the introduction (lines 46-47): “The Trichosporon genus has undergone extensive revision and the species T. beigelii has been replaced by several species”
- Illustration of results: Figure 3 A was supposed to illustrate a “granuloma” in the infected tussue. However, the photo included into the manuscript (3-A) just illustrates eosinophilic cellular debris without any clear resolution of the cell components that were described as “ mixed inflammatory cells in the dermis”. A granuloma is a chronic inflammatory process with a collection of cells of the mononuclear phagocyte system, including macrophages, in a typical tissue organization of the cell components. The photo used to illustrate a granuloma is not representative of the main cellular components required to define a granuloma. I suggest to replace this photo by another one more representative of a granulomatous inflammatory response.
- It is missing in the paper a discussion on the putative environmental source of contamination by Trichosporon in this particular captive animal who developed a disseminated fungal infection and fungemia by T. asahii. Finally, in humans, disseminated infection by T asahii are only documented in critically ill patients or patients with hematologic cancer. In this particular case report, I would expect to find an underlying condition that would predispose the animal to develop a disseminated infection by T asahii. The authors should comment on conditions usually associated to disseminated fungal infections in reptiles, with focus on the present report.
Author Response
Dear reviewers:
The attached file is revised manuscript with tracked changes (jof-1475686) and without tracked changes (jof-1475686-clean). In the revised manuscript, we have made the following changes according to the reviewers’ comments.
Reviewer-2’s comments
- Of note, all those modifications were not mentioned in any of the references listed by the authors in the present paper. This topic was extensively reviewed by Amir Arastehfar et al, in their publication at Crit Rev Microbiol 2021 47(6):679-698. I suggest to incorporate this reference right after the following sentence of the introduction (lines 46-47): “The Trichosporon genus has undergone extensive revision and the species T. beigelii has been replaced by several species”
Response: Thanks for the advice. We have added the reference “Arastehfar, A., de Almeida Junior, J.N., Perlin, D.S., Ilkit, M., Boekhout, T., Colombo, A.L., 2021. Multidrug-resistant Trichosporon species: underestimated fungal pathogens posing imminent threats in clinical settings. Crit Rev Microbiol 47, 679-698.”.
- Illustration of results: Figure 3 A was supposed to illustrate a “granuloma” in the infected tissue. However, the photo included into the manuscript (3-A) just illustrates eosinophilic cellular debris without any clear resolution of the cell components that were described as “ mixed inflammatory cells in the dermis”. A granuloma is a chronic inflammatory process with a collection of cells of the mononuclear phagocyte system, including macrophages, in a typical tissue organization of the cell components. The photo used to illustrate a granuloma is not representative of the main cellular components required to define a granuloma. I suggest to replace this photo by another one more representative of a granulomatous inflammatory response.
Response: Thanks for the advice. We have replaced fig 3a with a photo that has a higher magnification to illustrate the granuloma.
- It is missing in the paper a discussion on the putative environmental source of contamination by Trichosporon in this particular captive animal who developed a disseminated fungal infection and fungemia by T. asahii. Finally, in humans, disseminated infection by T asahii are only documented in critically ill patients or patients with hematologic cancer. In this particular case report, I would expect to find an underlying condition that would predispose the animal to develop a disseminated infection by T asahii. The authors should comment on conditions usually associated to disseminated fungal infections in reptiles, with focus on the present report.
Reponses: Thanks for the comment. The captive environment was established recently (i.e., it is a new facility), and there several possible factors that may compromise the immune function and lead to disseminated fungal infection, including transportation, environmental change, different housemates. For the source of T. asahii, since T. asahii is common in the environment, this lizard may be infected in the new facility. However, it is also possible that this lizard had been infected by T. asahii in previous facility (an old facility) as a subclinical/latent infection with subsequent disseminated infection in the new facility. We have included above discussion in the discussion section.
We thank you and the reviewers very much for providing us so many valuable suggestions. We hope the revise that we have made in the revision meets the reviewer’s and your requests. We are looking forward to having our revised manuscript published in your distinguished journal ‘‘Journal of Fungi”
Sincerely yours,
Wen-Ta Li, DVM, PhD
Pangolin International Biomedical Consultant Ltd.
Email: heerolee1104@gmail.com